# A qPCR assay for the detection of *Phytophthora abietivora,* an emerging pathogen on fir species cultivated as Christmas trees

**Guillaume Charron**[1]*, **Marie-Krystel Gauthier**[1], **Hervé van der Heyden**[2], **Guillaume J. Bilodeau**[3], **Philippe Tanguay**[1]*

**1** Natural Resources Canada, Centre de foresterie des Laurentides, Québec City, Canada, **2** Agriculture and Agri-Food Canada, Saint-Jean-sur-Richelieu Research and Development Centre, Québec, Canada, **3** Canadian Food Inspection Agency, Ottawa Plant Laboratory (Fallowfield), Ottawa, Ontario, Canada.

* guillaume.charron@nrcan-rncan.gc.ca (GC); philippe.tanguay@nrcan-rncan.gc.ca (PT)

## Abstract

Emerging species of the *Phytophthora* genus are among the most important threats to global plant biodiversity, horticulture and trade. For instance, *Phytophthora* root rot (PRR) of Christmas trees, mainly Fraser (*Abies fraseri*) and balsam (*Abies balsamea*) firs, is responsible for an average of 10% of the observed diseased trees in plantations. Diagnosing PRR involves isolation followed by morphological and molecular identification of the causal agents. However, these methods are rarely adapted to larger scale monitoring such as *in situ* detection. For these applications, molecular detection of environmental DNA (eDNA) provides the fast and high-throughput results needed. *Phytophthora abietivora* was associated with PRR in firs (*Abies spp.*) cultivated as Christmas trees in the province of Québec (Canada). This study focused on developing a sensitive and specific qPCR assay targeting *P. abietivora* and validating its efficiency on DNA extracted from soil and roots. A set of primers and probe was designed for this assay, and parameters such as the limit of detection ($LOD_{95\%}$) and limit of quantification (LOQ) were measured. The assay was tested on DNA obtained from healthy-looking and PRR symptomatic Fraser and balsam firs. The assay was shown to be semi-specific because it cross-reacted with *P. europaea* and three other phylogenetically close species, but deemed specific in the context of PRR of firs. The $LOD_{95\%}$ was estimated at 10 copies per reaction ($C_q$ of 35.7) and the LOQ to 33 *P. abietivora* oospores per gram of soil. Out of 488 DNA samples from soil, 68 were positive for *P. abietivora* from which 42 (61.7%) were from PRR symptomatic trees. Only a slight overlap (3 out of 7 samples) was observed with previously obtained baiting results. This assay will be useful for rapid diagnostics of *P. abietivora* infected trees and as a prospecting tool to better characterize the natural distribution and dissemination of the disease.

**Data availability statement:** All relevant data are within the manuscript and its Supporting Information files.

**Funding:** This study was supported by a grant in the Cellule d'innovation des méthodologies de diagnostic des ennemis de cultures (CIMDEC) from the Prime-Vert funding program of the Québec Ministry of Agriculture, Fisheries and Food (19-011-2.2-C-PHYTO) to H.H, P.T and G.J.B. https://www.mapaq.gouv.qc.ca/fr/Pages/Accueil.aspx The funders had no role in study design, data collection and analysis, decision to publish, or preparation of the manuscript.

**Competing interests:** The authors have declared that no competing interests exist.

## Introduction

Emerging plant pathogens have always threatened worldwide agriculture, natural ecosystems, and downstream human health [1,2]. Of these, species of the *Phytophthora* genus, with over 200 named species, represent the most destructive plant pathogens known and are responsible for historical economic losses [3–6]. For instance, one of the industries faced with *Phytophthora* root rot (PRR) is the cultivation of Christmas trees as ornamental plants, where PRR is one of the most prevalent and widely distributed diseases of true firs (genus *Abies*), killing about 10% of the planted Christmas trees in the United States of America [7]. Alone, PRR is responsible for millions of U.S. dollars in yearly losses for this industry, since infected trees are impossible to sell and die within months [8–10]. Disease symptoms such as root rot, wilting, and chlorotic foliage are hardly distinguishable from those caused by abiotic stresses and other soilborne pathogens such as *Pythium* and *Fusarium spp* [11].

Current diagnosis of PRR in firs requires laborious isolation and identification of the causal agent, which is further challenged given the secondary colonization of infected roots and the outcompeting growth of other fungi and oomycetes on semi-selective isolation media [12]. Direct molecular detection of *Phytophthora spp.* from diseased plant material and soil samples has relied lately on conventional PCR amplification of the internal transcribed spacer (ITS) of the ribosomal DNA (rDNA), nested PCR, followed by Sanger sequencing, and qPCR [13–16]. While detection is usually specific, the identification down to the species level can be hampered by the concomitant presence of multiple closely related *Phytophthora spp.* in the samples under investigation [17].

While *P. cinnamomi* is the main species causing PRR in the eastern United States [18], a recent survey identified *P. abietivora* as the species most frequently associated with dying balsam *(Abies balsamea)* and Fraser firs *(Abies fraseri)* cultivated as Christmas trees in Québec, Canada [19]. *Phytophthora abietivora* is a recently described oomycete pathogen isolated in a Connecticut fir plantation and confirmed as a causal agent of PRR in Fraser fir [20]. This species has also been found in the Pennsylvania Department of Agriculture's historical collection [18]. Because *P. abietivora* has been identified from samples pre-dating its formal description, Molnar et al. [18] suggested that due to the ITS sequence being highly conserved between *P. abietivora* and *P. europaea*, *P. abietivora* isolates could have been misidentified as *P. europaea* for decades. This high sequence identity for the ITS was also found between *P. abietivora* and four other species from clade 7a: *P. uliginosa*, *P. flexuosa*, *P. tyrrhenica* and *P.* sp. *cadmea* [21]. Bily et al. [15] reported associations between *P. abietivora* and tissues symptomatic for *Phytophthora* infections from multiple deciduous tree species. Based on these observations, *P. abietivora* could be endemic in North America and is now emerging as a pathogen with the potential to cause diseases in both deciduous and evergreen trees. Therefore, developing a molecular assay that detects *P. abietivora* in environmental samples or diseased tree samples is of utmost importance for both biomonitoring and diagnostic applications.

Specific identification and quantification of plant pathogens are now commonly achieved with reliable qPCR assays [12,22]. The high sensitivity makes it a valuable tool for early detection of pathogens, even before disease symptoms are visible [23]. As such, many studies have developed specific qPCR assays targeting *Phytophthora* species, such as *P. cryptogea* [24], *P. ramorum* [13], *P. cactorum* [25] and many others [26,27]. The need to develop an early and accurate molecular detection assay to monitor *P. abietivora* has become crucial, as treatment options are limited, and prevention remains the best option for growers.

The objectives of this study were to develop a sensitive and specific qPCR assay targeting *P. abietivora,* and to validate its efficiency on infected plant material and soil samples collected during our 2019-2021 survey of PRR in Christmas tree plantations in Québec, Canada.

## Materials and methods

### Quantitative PCR primers and probe design

The internal transcribed spacers (ITS) ITS1 and ITS2, including the 5.8S ribosomal RNA region of the rRNA cistron, was chosen as the target gene for our detection assay, as it is commonly used as a barcode for *Phytophthora* species. To design qPCR primers specific to *P. abietivora,* 14 sequences from various *Phytophthora* species isolated from Christmas tree plantations, including *P. abietivora*, were gathered from the GenBank database [19,28]. A sequence from an undescribed *Phytophthora* isolate obtained in a previous study [19] was also included. Fourteen sequences from other *Phytophthora* never reported in association with fir trees were also added, three of which (*P. uliginosa*, *P. flexuosa* and *P. tyrrhenica*) are part of an unresolvable genetic cluster for the ITS1 sequence along with *P. abietivora* and *P. europaea*. The sequences were aligned using Clustal X [29], and the alignment was reviewed for variations to identify potential target regions. Initial specificity assessment was made by querying the GenBank database using the BLAST Sequence Analysis Tool [30] with the sequences of the selected primers and probe (S1 Table).

### Real-time qPCR conditions for the P_abi_detect assay

A QuantStudio 5 instrument (Thermo Fisher Scientific, Waltham, United States) was used to determine the specificity and sensitivity of the P_abi detect qPCR assay. All the qPCR reactions in this study were run on a QuantStudio 5 or a 7500 Fast qPCR (Thermo Fisher Scientific, Waltham, United States) instruments. Primers and probe sequences are listed in Table 1 and were synthesized by Integrated DNA Technologies Inc. (Coralville, IA, USA). Cycling conditions were as follows: 5 minutes at 95°C followed by 40 cycles of 15 seconds at 95°C and 45 seconds at 60°C. A single reaction mix comprised 1× SensiFAST™ Probe No-ROX mix (Bioline Meridian Bioscience, Cincinnati, Ohio, United States), 500 nM of P_abi_ITS_F and P_abi_ITS_R primers, and 100 nM of P_abi_ITS_probe. Unless stated otherwise, the total reaction volume for a standard P_abi_detect detection assay was 10 µL.

### Assay specificity

The specificity of the assay was tested with the genomic DNA from 24 of the 29 species used in the alignment for primer design (no DNA was available for *P. flexuosa, P. uliginosa, P. tyrrhenica, P. pseudotsugae* and *P. citrophthora*). The DNA was obtained from existing in-house collections or was shared by collaborators (S2 Table). Some DNA samples had known DNA concentrations either quantified with a NanoDrop ND-1000 Spectrophotometer instrument (Thermo Fisher Scientific, Waltham, United States) or a Qubit dsDNA assay kit (Thermo Fisher Scientific, Waltham, United States). These samples were diluted to 100 pg/µL for the specificity assay. Other samples had a known concentration in copies (5,000) of Beta-tubulin determined by SYBRGreen real-time PCR quantification [31]. These samples were diluted to 500 copies/µL. For each sample, the assay was run in triplicate on a single 96-well plate. DNA extracted from a purified strain of *P. abietivora* was used as a positive control and PCR-grade

**Table 1. Primers and probe used in the *P. abietivora* (P_abi_detect) detection assay.**

| Primer/Probe | Sequence | Length (bp) | $T_m$ (°C) | Amplicon length (bp) |
|---|---|---|---|---|
| P_abi_ITS_F | 5'-CCCACAGTATATTCAGTATTCAA-3' | 23 | 55.0 | 181 |
| P_abi_ITS_R | 5'-TGAACCGTATCAACCCA-3' | 17 | 53.8 | |
| P_abi_ITS_Probe | 5'-/56-FAM/AGCCACCAG/ZEN/GACAAGC/3IABkFQ/-3' | 16 | 60.0 | |

water was used as a negative control [19]. To be considered positive, a qPCR reaction had to generate an amplification curve with a $C_q$ lower than the cut-off values determined with the $LOD_{95\%}$ in all three replicates, as detailed in the results section.

In addition to the panel of species, a panel of 10 *P. abietivora* isolates and 21 *P. europaea* isolates was also tested (S3 Table). The DNA from *P. abietivora* isolates was extracted from purified mycelium, while the DNA from *P. europaea* isolates was extracted from a small quantity of mycelium on agar plugs using the protocol developed in Penouilh-Suzette et al. [32] with minor modifications as in Charron et al. [19]. The DNA extracts concentrations were measured with a NanoDrop ND-1000 Spectrophotometer (Thermo Fisher Scientific, Waltham, United States). The extracts were diluted to 100 pg/μL, of which 1 μL was used in three technical replicates of the assay. For isolates with failed amplification at 100 pg/μL, the assay was rerun with 1 μL at 500 pg/μL.

### Limit of detection (LOD)

To assess the 95% limit of detection ($LOD_{95\%}$) of the assay, a gBlock™ (Integrated DNA Technology Inc., Coralville, IA, USA), including the 181 bp of our target sequence with 24 bp flanking regions on each side, for a total of 229 bp, was designed (S4 Table). The copy number of the gBlock™ stock solution was estimated using the following equation:

$$number\ of\ copies\ (molecules) = \frac{x \times N_A}{(y \times 141{,}375.3\ g/mole) \times 10^9\ ng/g}$$

Where x is the amount of the *P. abietivora* ITS2 gBlock™ in nanograms, $N_A$ is the Avogadro constant ($6.0221 \times 10^{23}$), and y is the length of the *P. abietivora* ITS2 gBlock™ in bp.

A solution at 10 million copies per microliter was prepared and diluted tenfold seven times to obtain $10^6$ to 1 copy per microliter dilutions. One microliter of each dilution was used in three technical replicates of the assay. Because the 1 and 10 copies dilutions had great variations in their $C_q$ values, only the dilutions from 100 copies to $10^6$ copies were used to calculate the standard curve parameters. The standard curve equation was determined by linear regression analysis using the $Log_{10}$ number of copies as the predictor variable, and the $C_q$ value as the response variable. The efficiency of the amplification was calculated as:

$$Efficiency (\%) = 1 - 10^{(-1/slope)} \times 100$$

Once the standard curve was established, the $LOD_{95\%}$ was tested with twofold serial dilutions of the gBlock™, including 20, 10, 5, 2.5, 1.25, and 0.625 copies per microliter. For each dilution, 5 μL was used in 20 technical replicates of the assay (100, 50, 25, 12.5 6.25 and 3.1 copies per reaction). A successful amplification was defined as any amplification with a $C_q$ value under 40. The theoretical $LOD_{95\%}$ was determined with a binary logistic regression (probit model) analysis using the $Log_{10}$ of the number of copies as the predictor variable, and the positive/negative detection result as the response variable. The minimal number of copies necessary to obtain a positive result in 95% of the assays ($LOD_{95\%}$) was determined with the resulting binary logistic regression equation. The $LOD_{95\%}$ value was empirically verified by making a dilution of the gBlock™ to the determined $LOD_{95\%}$ value and tested in 20 replicates.

### Limit of quantification (LOQ)

To estimate the limit for *P. abietivora* quantification in soil samples, oospore suspensions of different concentrations were prepared. The *P. abietivora* isolate CFL5946 (isolate 062_E in Charron et al 2024 [19]) was inoculated on a V8-Phytagel medium (5% V8 juice, 0.4%

Phytagel™ (MilliporeSigma, Burlington, USA) amended with 10% of a macro-element solution (10 g/L KH$_2$PO$_4$, 5 g/L MgSO$_4$•7H$_2$O and 1 g/L CaCl$_2$•2H$_2$O) and incubated at 20°C. After 7 days, the mycelium obtained was used to inoculate five new V8-Phytagel™ media that were incubated for another 7 days at 20°C. Following incubation, the Phytagel™ media were dissolved in 1L of citrate buffer (10 mM, pH 6) and filtered on 100 μM nylon mesh to gather the *P. abietivora* mycelium. To free the oospores, the collected mycelium was resuspended in 30 mL of an aqueous enzyme solution containing 10 mg/mL Driselase™ Basidiomycetes sp. (MilliporeSigma, Burlington, USA) and 6.7 mg/mL of lysing enzymes from *Trichoderma harzianum* (Millipore Sigma, Burlington, USA) and incubated overnight at 30°C on a tube rotator (VWR International, Radnor, USA). Complete digestion of the mycelium was confirmed by microscopic observation. The oospore suspension was centrifuged 5 minutes at 2800 × g, and the supernatant was removed. The oospore pellet was washed twice in phosphate-buffered saline (PBS, pH 7.4), and resuspended in 1 mL of PBS. Oospore count was estimated using an Improved Neubauer hemacytometer (American Optical Company, Buffalo, New York). This oospore stock (≈830 cells/μL) was used to prepare suspensions of known total oospore quantity (8300, 830, 83, 8.3, 6, 4, 2, 1, and 0.5 oospores). For each oospore quantity, total DNA extractions were performed on oospores alone or mixed with soil in four replicates.

For DNA extraction of oospores alone, the oospores were resuspended in 180 μL of ATL buffer from the QIAamp DNA mini kit (Qiagen, Valencia, CA, USA) and DNA was extracted according to the manufacturer's specifications. For DNA extractions of oospores mixed with soil, the oospores were added to a PowerBead pro tube containing 250 mg of garden soil free from *P. abietivora,* and DNA was extracted following the DNeasy® Powersoil Pro kit protocol (QIAGEN, Hilden, Germany). The soil was taken from a lot of the community garden at the Laurentian Forestry Center (46.779, -71.282). To make sure this soil was not previously contaminated with *P. abietivora*, six independent samples were used for DNA extractions with the Powersoil Pro protocol. On each of the DNA samples obtained, the detection assay was performed in three technical replicates.

In both extractions on oospores alone and on oospores in soil, elution was performed in 100 μL of PCR grade water. Following extraction, 1 μL of each DNA sample was used with the *P. abietivora* detection assay in three technical replicates, providing data for DNA equivalent of 2 to about 33,000 oospores per gram of soil. After data compilation, the Hampel filter method was used to identify and remove potential outlier values from the final data set [33]. This filter considers as outliers the values outside the interval formed by the median, plus or minus 3 median absolute deviations. The limit of quantification was established as the lowest number of oospores, with a 100% positive detection rate in the replicates left after outlier removal.

## Soil and tree samples preparation

Soil associated with trees of three different statuses (PRR-like symptomatic trees from Christmas tree plantation, asymptomatic trees from the same plantation, and trees from natural forest nearby the plantation) were sampled in a previous study [19] from 40 sites across the Estrie and Chaudière-Appalaches regions (Québec, Canada) between 2019 and 2021. In short, at each site, three soil subsamples 120° apart were taken from the crown limit at a 17-cm depth with a soil auger. The organic horizon was removed from the three soil cores and each of them was cut in half lengthwise, where one half of each core was discarded. The remaining halves were pooled and formed the final sample [19]. Unidentified root fragments were sampled only from the trees showing PRR-like symptoms. The soil and root samples were prepared as described in Charron et al [19]. In summary, the soil samples were homogenized by sieving

them through a 4-mm mesh basket. From this homogenized soil, approximately 250 mg were transferred to a 2 mL tube for DNA extractions. Root fragments retained in the sieve were cut into 5 to 10-mm sections with a sterile scalpel and transferred to a 2 mL tube for DNA extractions. Prior to DNA extractions, root samples were kept at −20°C between 3 months (2020-2021 samples) up to 2 years (2019 samples) and soil samples were kept at −20°C for up to 2 months.

## DNA extractions

DNA from soil and root fragments (Estrie and Chaudière-Appalaches) was extracted with the DNeasy® Powersoil kit (QIAGEN, Hilden, Germany, discontinued) with slight modifications. Instead of 10 minutes, the samples were vortexed to maximum speed for 20 minutes. PCR grade water was used for the elution of DNA instead of the C6 Buffer. For each DNA sample obtained from soil and root fragments, 1 μL was tested for the presence of *P. abietivora* with the P_abi_detect assay in three technical replicates.

## Data visualization and statistical analyses

All data visualization was made with the help of the R software v4.1.3 [34] and custom scripts. Unless stated otherwise, the significance level used for all statistical tests is α = 0.01 and p-values were adjusted with the false discovery rate method when appropriate (fdr) [35]. Maps were generated with the package sf v1.0-16 [36] and SHP files publicly available from the Geographic and administrative database [37]. Linear and binary logistic regressions were respectively performed with the lm and glm functions of the R software v4.1.3 [34]. Phi coefficient for the comparison of metagenomic data with the detection data were calculated with the phi function of the psych v2.4.6 [38].

# Results

## Quantitative PCR design

The target region of our P_abi_detect assay can be found in the ITS2 sequence between position 728 and 917 of our alignment (S1 Table). While the sequences of the primers P_abi_ITS_F and P_abi_ITS_R do not prevent the amplification of nonspecific targets, the assay's specificity comes from the probe P_abi_ITS_Probe (S1 Table and Table 1). However, the two sister species *P. europaea* and *P. abietivora,* as well as *P. flexuosa, P. tyrrhenica,* and *P. uliginosa* all have 100% sequence identity in the target region. The next closest species to *P. abietivora* in our alignment, *P. fragariae*, has two SNPs in the sequence targeted by the probe (S1 Table).

## Assay specificity

Criteria for positivity were first established as follows: to be considered positive, a sample had to produce an amplification curve for all three replicates, with a $C_q$ lower than 35.7, a value determined by the experimental $LOD_{95\%}$ (see section on LOD). As expected, the P_abi_detect qPCR showed specific amplification when 100 pg genomic DNA from *P. abietivora* or *P. europaea* was used in tests (Fig 1A). No nonspecific amplification under the $LOD_{95\%}$ cut-off was observed in the tested conditions for the genomic DNA of the 22 other species tested, including members of the subclades 7a and 7c (*P. fragariae*, *P. × cambivora*, and *P. cinnamomi*). Following the criteria for positivity established above, three species (*P. cactorum*, *P. sansomeana* and the unknown *Phytophthora*) amplified at a late cycle, past the cut-off and for only for one replicate out of three, were therefore not considered as positive and may be the result of a cross-contamination.

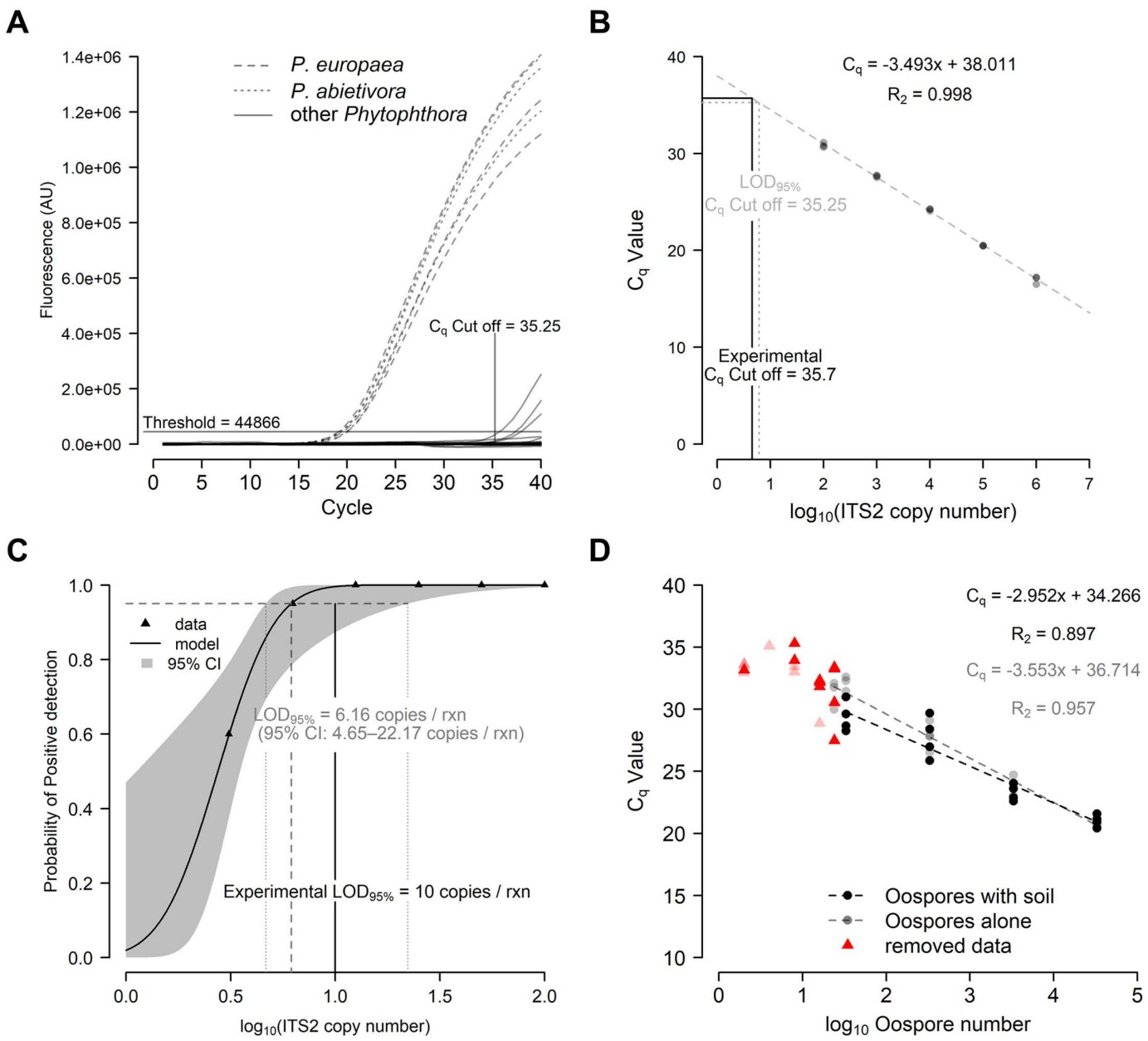

**Fig 1. The P_abi_detect assay is specific and sensitive.** (A) The P_abi_detect qPCR assay showed specific amplification from *P. abietivora* and *P. europaea* genomic DNA while DNA from the other species tested do not result in reliable amplification under the established cut-off. (B) The P_abi_detect assay showed a linear response between $C_q$ values and the ITS2 copy numbers. (C) The Probit regression analysis used to estimate the LOD95% of the assay. The dashed lines intercept at the theoretical 95%. (D) The P_abi_detect shows a linear response between $C_q$ values and the quantity of oospores present in samples spiked with known quantities of cells.

From the DNA of 31 independent *P. abietivora* and *P. europaea* isolates tested, 29 (nine *P. abietivora* and 20 *P. europaea*) consistently tested positive with an average $C_q$ value of 23.1 (S3 Table). The variation in $C_q$ between samples was high, which could be due to the initial dosage of DNA. Two DNA samples, one *P. abietivora* and one *P. europaea*, produced less than three positive replicates and were retested to verify if this was due to an overestimation of the DNA concentration due to poor quality samples. The samples were retested at 500 pg of DNA per

reaction. The assay gave a clear positive signal for the *P. europaea* DNA samples (AX44). The *P. abietivora* sample (12324_2CR) also failed to produce a positive detection at 500 pg. The identity of the species was verified by Sanger sequencing of the whole ITS1-5.8S-ITS2 region. A BLAST of the obtained sequence against the nt/nr database of the NCBI confirmed a misidentification as the DNA did not match to *P. abietivora*, but *P. gonapodyides*.

## Standard curve

Our assay has shown a linear response between 1 million target gBlock™ copies down to 100 target gBlock™ copies per reaction (Fig 1B). The amplification efficiency of 93.3% suggests that PCR products are not exactly doubling at each cycle. However, this value is within the desired efficiency range of 90 to 110% [39,40].

## Limit of detection (LOD)

We evaluated the analytical sensitivity of the P_abi_detect qPCR with two-fold dilutions of a positive control gBlock™ between 100 and 3.1 copies per reaction. The binary logistic regression analysis estimated the $LOD_{95\%}$ at 6.16 copies per reaction (Fig 1C). The 95% confidence interval was from 4.65 to 22.17 copies per reaction. Repeating the P_abi_detect qPCR assay at 6.16 copies per reaction in 20 replicates yielded only a 75% positive rate. To provide a more reliable analytical cut-off for assay positivity, we tested 8,10, and 25 copies per reaction of the control gBlock™. At 10 copies per reaction, the assay returned a 95% positive rate with an average $C_q$ of 35.7 (S5 Table). Extrapolating the $C_q$ value from the copy number with the regression line of our standard curve would result in an analytical cut-off of 34.5. However, since these values fall outside the linear part of the standard curve, that 10 copies represent a quantity within the 95% confidence interval, and that, in a disease management context, false negatives are more detrimental than false positives, we decided to use the mean $C_q$ value obtained for 10 copies per reaction as experimentally tested. This resulted in a more conservative analytical $LOD_{95\%}$ of 35.7. In all subsequent experiments, $C_q$ values exceeding this cut-off would be considered negative.

## Limit of quantification (LOQ)

We estimated the limit of quantification by adding different suspensions of known oospore quantities, ranging from 0.5 to about 33,000 oospores, in DNA extractions with soil and without soil. A control experiment conducted on the garden soil used for LOQ determination revealed that neither *P. abietivora*, nor any species from the *P. europaea* species complex, were present before the addition of the oospores, as all six soil samples returned negative detection results.

After outlier removal, multiple linear regression was used to test if the quantity of oospores present in a sample significantly predicted the assay $C_q$ values for DNA extracted from soil spiked with oospores or oospores alone (Fig 1D). The overall regression was statistically significant ($R^2 = 0.937$ (multiple) and 0.931 (adjusted), $F(3,30) = 148.3$, $p < 0.001$). Both the $Log_{10}$ of the oospore quantity in soil ($\beta = -2.952$, $p < 0.001$) and pure oospores ($\beta = -3.553$, $p < 0.001$) DNA extracts significantly predicted the $C_q$ values. While there was a significant difference between the two intercepts (2.448, $p = 0.021$), the difference in slope was not significant (-0.601, $p = 0.07$). The lowest number of oospores detected in 100% of the tests was 8.3 or about 33 oospores per gram of soil. The average $C_q$ values for this oospore quantity were 29.4 (with soil) and 32.1 (only oospores).

## Case study from plantations

The P_abi_detect assay was used to detect *P. abietivora* from DNA extracted from soil samples taken under 1) healthy-looking firs in natural forest, 2) healthy-looking firs in plantation,

and 3) PRR symptomatic firs in plantation (Fig 2). Unidentified root samples were only taken from the rhizosphere soil of PRR symptomatic firs. Of 488 soil samples, 68 were declared positive for *P. abietivora.* Out of the 68, 42 were from PRR symptomatic trees, 17 from healthy trees within the plantations and 9 from trees from natural forest stands near the plantations (Fig 3A). While the low number of strains isolated by baiting preclude any statistical analysis, only a slight overlap was found between the results obtained by baiting and results from the detection of *P. abietivora* DNA in the soil (3 out of 7 samples). No statistically significant difference in the proportion of positive detection was found between *Abies* species sampled (2-sample test for equality of proportion, $p = 0.013$), region sampled (2-sample test for equality of proportion, $p = 0.069$) or years of sampling [Pairwise comparison of proportions, $p = 0.065$ (2020-2021 and 2019-2020) and $p = 0.953$ (2019-2021)]. However, we found a significant difference in the proportion of positive detection between soil sampled from PRR-like symptomatic trees and healthy trees [Pairwise comparison of proportions, $p < 0.001$ (PRR-healthy forest), $p < 0.001$ (PRR-healthy plantation), and $p = 0.172$ (healthy forest-healthy plantation)] (Fig 3A). Accordingly, a binary logistic regression using the result of the assay (positive, negative) as the dependent variable showed that soil DNA samples from PRR symptomatic trees in Christmas tree plantations had higher odds of being positive for *P. abietivora* (OR = 5.96; 95% CI: 2.77 to 12.85; $p < 0.001$, Fig 3B) compared to healthy looking trees from natural forests. The odds of obtaining a positive detection result from healthy-looking trees in Christmas tree plantations were also higher than healthy-looking trees from natural forests, but this difference was not statistically significant (OR = 1.92; 95% CI: 0.82 to 4.48; $p = 0.13$, Fig 3B). For PRR symptomatic trees, the proportion of positive detection did not significantly differ between soil (42/164) or root (28/162) samples (2-sample test for equality of proportions with continuity correction, $X^2(1) = 1.79$, $p = 0.09$). There were only 16 samples overlapping for positive detection between both sample types (Fig 4). The average $C_q$ values obtained for soil and root samples were also not significantly different (Welch two sample t-test, t (50.19) = 1.4911, $p = 0.14$).

A subsample of 316 soil samples was used for a metagenomic analysis by Van der Heyden *et al.* [21] and overlaps with the samples tested in this study. Overall, for 89.2% (282/316) of soil samples, both methods were in agreement on the presence/absence of *P. abietivora*. Accordingly, the *P. abietivora* detection data shows a positive relationship with the presence of reads from the *P. europaea* species complex for any given soil sample (Phi coefficient: 0.65, Fisher's exact test corrected $p < 0.001$). While the relationship is weaker in healthy samples from plantations (Phi coefficient: 0.59, Fisher's exact test corrected $p < 0.001$) and healthy trees in natural forest near plantations (Phi coefficient: 0.51, Fisher's exact test corrected $p < 0.001$), it is the strongest when comparing data from PRR symptomatic trees (Phi coefficient: 0.74, Fisher's exact test $p < 0.001$).

## Discussion

Since *P. abietivora* is a recently described species [20], most of the information about its distribution and host range remains to be discovered. To facilitate and speed up filling these gaps in knowledge, we developed a sensitive and semi-specific qPCR assay to detect *P. abietivora* in environmental samples (diseased plant tissues and soil).

Although the P_abi_detect assay was not designed with quantification in mind, the parameters obtained with the standard curve indicate that the assay can be used in a quantitative manner. Based on the data obtained for gBlock™ as the target sequence, as little as 10 copies of the target sequence should be detectable. While this information was not obtained for *P. abietivora* genomic DNA, the results obtained from the assay, when used on the DNA of different strains, suggest that it is as sensitive as other TaqMan assays developed in the ITS for

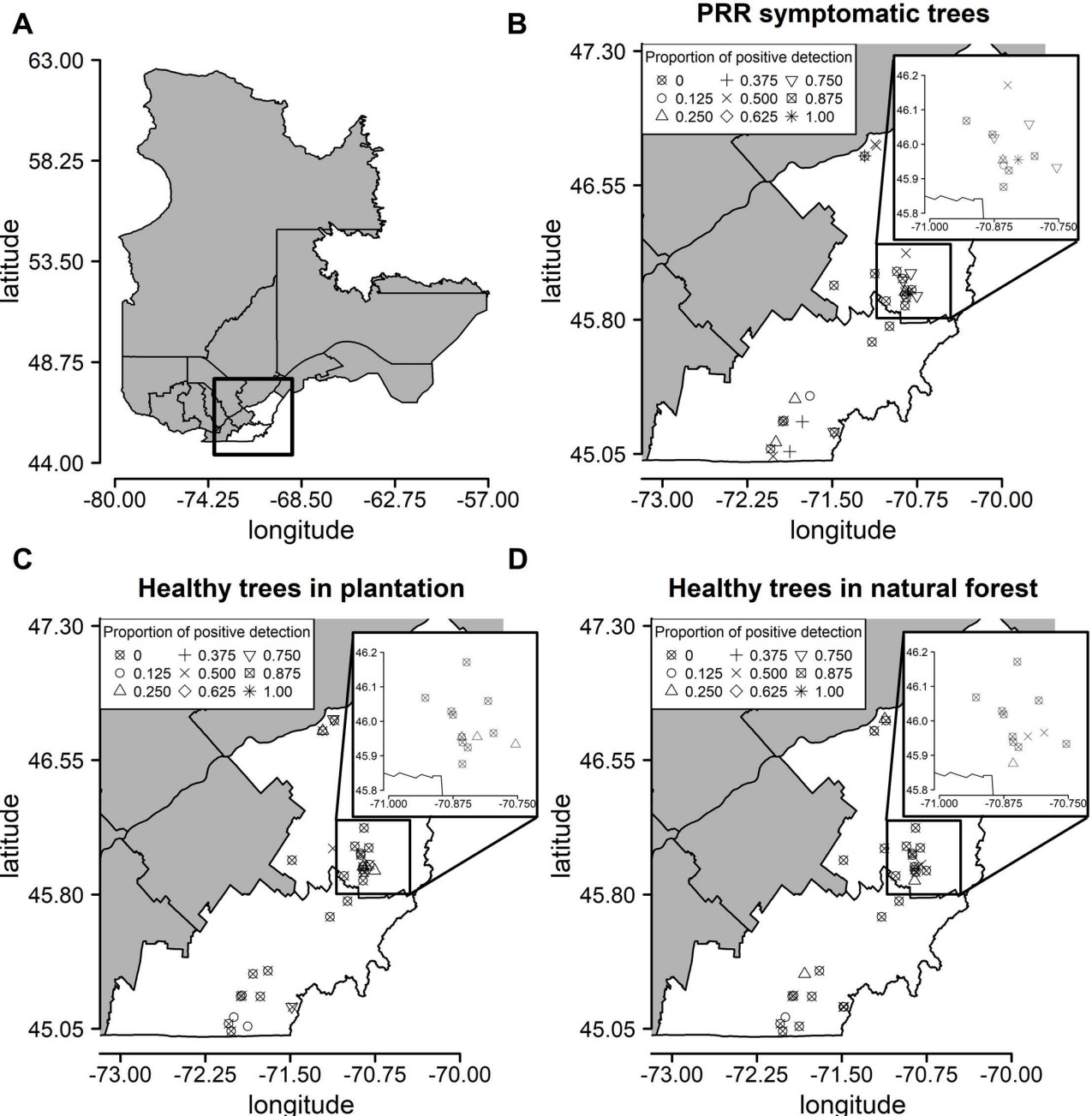

**Fig 2. *Phytophthora abietivora* is present in major Christmas tree producing regions of the province of Québec.** (A) Map of the province of Québec with the regions of Estrie (Southward) and Chaudière-Appalaches (Northward) highlighted in white. (B) Proportion of positive detections in soil samples of PRR symptomatic firs trees for each sampled site. (C) Proportion of positive detections in soil samples of healthy-looking firs trees for each sampled site. (D) Proportion of positive detections in soil samples of healthy-looking firs trees in natural forest stands in the vicinity of each sampled site. The insets in (B), (C), and (D) represent an enlarged region with a high density of sampled Christmas tree productions. The data used to create these maps was obtained from the Ministry of Natural Resources and Forests under a CC BY 4.0 license, with permission from MNRF, original copyright 2024. [https://www.donneesquebec.ca/recherche/dataset/base-de-donnees-geographiques-et-administratives] [37].

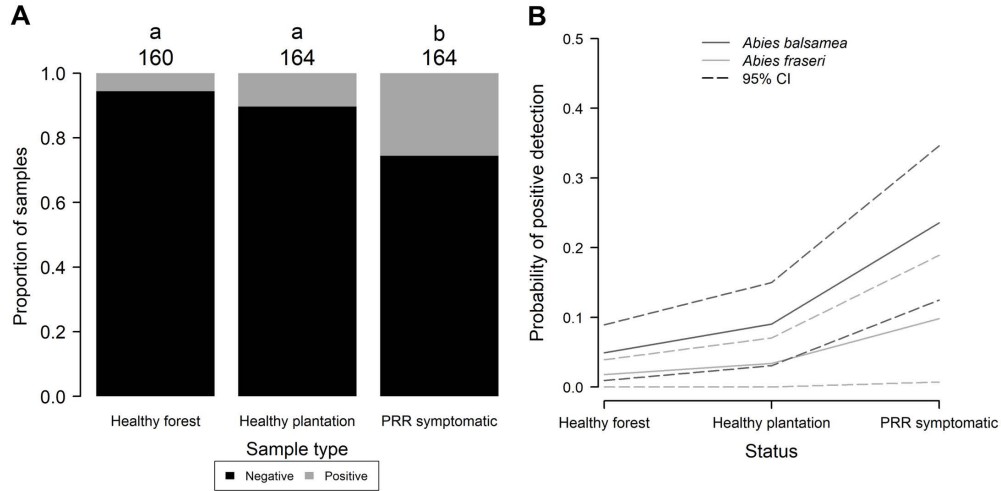

**Fig 3. PRR-like symptoms are a predictor of the presence of *P. abietivora* in the associated rhizosphere soil.**
(A) Bar plot showing the proportion of positive detections for the soil samples of different tree status. The number of samples in each category is indicated above the bars. (B) Binary logistic regression using the result of the assay (positive, negative) as the dependent variable. For both cultivated *Abies* species, samples taken in the rhizosphere soil of PRR-like symptomatic trees have higher odds of being positive for the presence of *P. abietivora* than the other tree status.

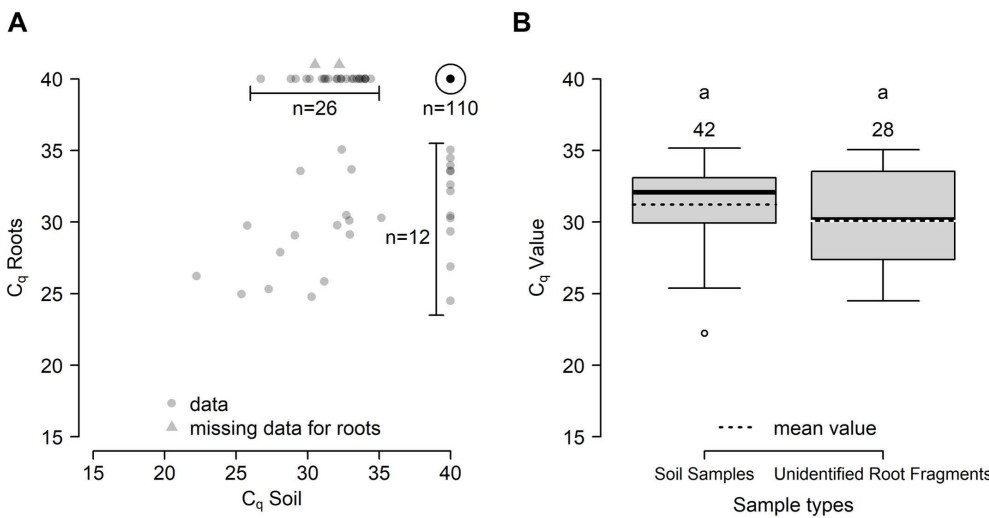

**Fig 4. DNA from *P. abietivora* is found in soil and root fragments samples in comparable quantities.** (A) Scatter plot of $C_q$ values for the 164 samples from PRR-like symptomatic trees. (B) Boxplot of the $C_q$ values obtained for positive soil and unidentified root fragments samples. Number of samples is displayed above the individual plots.

other *Phytophthora* species [12]. Our assay takes advantage of the high copy number of the rRNA in the *Phytophthora* genomes [41]. This increases the sensitivity of our assay in samples with low pathogen concentration compared to other assays based on the amplification of single-copy target genes. Even in the case of mature oospores, which contain only one diploid nucleus [42], our assay provided detection at concentrations as low as 33 oospores per gram of soil. Previous studies have shown that the detection of *Phytophthora* spp. DNA from soil is feasible even at those low densities [43,44], while other found it harder to effectively detect

*Phytophthora* pathogens due to DNA extraction limitations [45]. Latent structures such as oospores from *P. capsici* were still viable after two years in soil exposed to natural environmental conditions [46], and for at least five years for *P. abietivora* [20].

In the past, others have targeted different genes to differentiate *Phytophthora* species [47], such as Ypt1 [48], Atp9-Nad9 [26,27] or Rps10 for oomycetes in general [49]. While using the ITS as a target for detection offers advantages towards sensitivity, it does come with a trade-off in specificity. The ITS as a marker provides resolution between distant species, but its use is sometimes challenged when dealing with closely related species [50]. For the P_abi_detect assay, this translates as being only semi-specific as the ITS2 sequence targeted by the primers and probe is identical between the five members of the *P. europaea* species complex, which includes *P. abietivora*, *P. flexuosa*, *P. uliginosa* and *P. tyrrhenica* [21]. This is not uncommon as other assays targeting *Phytopthora* pathogens have also showed cross-reactivity with other closely related species (*P. cactorum* [25], *P. cinnamomi* [51], *P. infestans* [45,52]). In those cases, the validity of the assay is preserved as species cross-reacting with the assay are often not found in the same environments as the targeted pathogen. For the P_abi detect assay, this shortcoming is mitigated by the fact that, besides *P. europaea*, none of the other members of this complex has ever been reported in association with evergreen trees or reported in North America (*P. flexuosa*) [53], (*P. uliginosa*) [54], (*P. tyrrhenica*) [55]. Previous reports of *P. europaea* on North American firs are questionable since there are chances that they resulted from misidentification due to the high identity between the ITS sequences of *P. europaea* and *P. abietivora*. In cases for which isolates are available, sequencing of loci offering more discrimination will be needed for confirmation. The P_abi_detect assay has been developed to provide a tool for detection and diagnosis in North American Christmas tree productions. Remaining within this context excludes the presence of the other species from the *P. europaea* species complex. When the presence of other species from this complex cannot be excluded, this assay should always be coupled with confirmation methods such as strain baiting or strain isolation from diseased tissues, followed by Sanger sequencing of a discriminating locus to confirm the presence of only *P. abietivora*. There is currently limited genetic information for *P. abietivora*, with only 6 different loci sequences available from the GenBank® database. While these sequences offer enough resolution for taxonomic studies, developing specific detection assays will require more sequence information. Whole genome sequencing of *P. abietivora* and the other species from the complex will undoubtedly facilitate the development of more specific detection tools.

We used the P_abi_detect assay on DNA extracted from different samples gathered in Christmas tree plantations during a previous study [19]. In this study, *P. abietivora* was identified as a potential pathogen, with eight strains being isolated from independent PRR-like symptomatic tree samples. Based on isolation data, *P. abietivora* was detected in 4.3% (7 out of 164) of the rhizosphere soil samples and 0.6% (1 out of 164) of the root samples [19]. In contrast, with the P_abi_detect assay, positive detections for rhizosphere soil samples and roots samples increased 5.95-fold (to 25.6%) and 28-fold (to 17%), respectively. These results further reinforce the hypothesis of a causal link between *P. abietivora* and the PRR diseases observed in the Québec Christmas tree plantations. Moreover, we detected *P. abietivora* in the rhizosphere soil of healthy-looking trees from both the plantations (10.4% of samples) and nearby natural forests (5.6% of samples), whereas none of those samples yielded any *P. abietivora* isolates.

A recent study by Van der Heyden et al [21], built on a subset of the samples analyzed in the present study, used a metagenomic approach and showed the presence of the *P. europaea* complex (most probably *P. abietivora*) in 29.9% of the soil samples collected under trees showing PRR-like symptoms. They also showed that 15% of the soil samples collected under

healthy trees also presented reads from the complex. As expected, there is a strong positive association between the results of the P_abi_detect assay and the presence of reads from the *P. europaea* species complex for a given sample.

These independent methods both confirm that *P. abietivora* is present in natural forest stands neighboring the plantations. Its association with the rhizosphere soil of balsam firs from natural forests suggests that it could either be a native pathogen of this fir species or that it spread from a nearby infected Christmas tree plantation. Since little is known about the current distribution of *P. abietivora*, our detection assay can be used in mapping efforts to establish its presence in diverse environments such as nurseries, Christmas tree farms, and natural forests.

Large differences in sensitivity between baiting and molecular detection of *Phytophthora* species in soil samples are not uncommon. Using a nested PCR, *P. cinnamomi* was detected in 96% of the soil samples taken at the margin of the disease front, while only 2.2% of the same samples yielded isolates from baiting [56]. This is not surprising as successful baiting relies on the formation of sporangia, release of zoospores from the sporangia, and infection of the bait tissue by the zoospores. However, various microorganisms such as bacteria and fungi present in the soil can potentially interfere with the growth, reproduction, and infectivity of *Phytophthora* species by producing compounds that can inhibit the production of zoospores [57]. While more sensitive, using DNA based molecular detection only indicates the presence of the pathogen without any information about its viability status. A positive detection with molecular methods does not always equate to living cells in the sample as it was shown that DNA can persist through time in the environment following the death of organisms [58,59]. It was estimated that the DNA of *P. cinnamomi* cells can persist for more than a year in soil [51]. In consequence, culturing should be used in conjunction with our assay to confirm the viability of the pathogens detected. As demonstrated for *P. ramorum*, our assay should also be tested on reverse transcribed RNA to assess its utility to detect and quantify viable propagules of *P. abietivora* [60]. Such an assay could then be used to estimate the reduction of *P. abietivora* propagules in soil following disinfection treatments.

In conclusion, this qPCR assay will prove useful in the early detection and biomonitoring of *P. abietivora* in Christmas trees plantations, hence saving this industry from several millions of dollars in losses. Furthermore, it will facilitate the gathering of crucial information on both the distribution of this species in nature and its host range, leading to a better understanding of the ecology of this emerging pathogen.

## Supporting information

**S1 Table. Alignment of the targeted sequences for different *Phytophthora* species.**
(XLSX)

**S2 Table. DNA from *Phytophthora* isolates used to assess the specificity of the P_abi_ detect assay.**
(XLSX)

**S3 Table. DNA from *P. abietivora* and *P. europaea* isolates used to test the P_abi_detect assay.**
(XLSX)

**S4 Table. Sequence of the gBlock™ fragment used for the standard curve of the P_abi_ detect assay.**
(XLSX)

**S5 Table. Data obtained for the LOD$_{95\%}$ experimental validation.**
(XLSX)

**S1 File. Data and script used for analyses.**
(ZIP)

## Acknowledgments

We would like to thank Amélie Potvin and Nathan Benoît for their technical assistance and insightful discussions; Christopher Keeling for his comments on the manuscript; Jean-François Légaré, Éric Dussault, Christian Lacroix, Dominique Choquette and Jacinthe Drouin for their contributions to fieldwork and sampling; and all the growers for giving us access to their plantations.

## Author contributions

**Conceptualization:** Hervé Van der Heyden, Guillaume J. Bilodeau, Philippe Tanguay.

**Data curation:** Guillaume Charron, Marie-Krystel Gauthier.

**Formal analysis:** Guillaume Charron.

**Funding acquisition:** Hervé Van der Heyden, Guillaume J. Bilodeau, Philippe Tanguay.

**Investigation:** Guillaume Charron.

**Project administration:** Hervé Van der Heyden, Guillaume J. Bilodeau, Philippe Tanguay.

**Validation:** Guillaume Charron, Marie-Krystel Gauthier.

**Visualization:** Guillaume Charron, Marie-Krystel Gauthier.

**Writing – original draft:** Guillaume Charron, Marie-Krystel Gauthier, Philippe Tanguay.

**Writing – review & editing:** Guillaume Charron, Marie-Krystel Gauthier, Hervé Van der Heyden, Guillaume J. Bilodeau, Philippe Tanguay.

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
