## [Decision Letter · Decision Letter 0]

20 Nov 2024

PONE-D-24-40058A qPCR assay for the detection of *Phytophthora abietivora* , an emerging pathogen on fir species cultivated as Christmas treesPLOS ONE

Dear Dr. Charron,

Thank you for submitting your manuscript to PLOS ONE. After careful consideration, we feel that it has merit but does not fully meet PLOS ONE’s publication criteria as it currently stands. Therefore, we invite you to submit a revised version of the manuscript that addresses the points raised during the review process.

We look forward to receiving your revised manuscript.

Kind regards,

Kandasamy Ulaganathan

Academic Editor

PLOS ONE

“This study was supported by a grant in the Cellule d’innovation des méthodologies de diagnostic des ennemis de cultures from the Prime-Vert funding program of the Québec Ministry of Agriculture, Fisheries and Food (19-011-2.2-C-PHYTO). https://www.mapaq.gouv.qc.ca/fr/Pages/Accueil.aspx”

3. We note that Figure 2 in your submission contain [map/satellite] images which may be copyrighted. All PLOS content is published under the Creative Commons Attribution License (CC BY 4.0), which means that the manuscript, images, and Supporting Information files will be freely available online, and any third party is permitted to access, download, copy, distribute, and use these materials in any way, even commercially, with proper attribution. For these reasons, we cannot publish previously copyrighted maps or satellite images created using proprietary data, such as Google software (Google Maps, Street View, and Earth). For more information, see our copyright guidelines: http://journals.plos.org/plosone/s/licenses-and-copyright.

Reviewers' comments:

Reviewer's Responses to Questions

**Comments to the Author**

1. Is the manuscript technically sound, and do the data support the conclusions?

Reviewer #1: Yes

Reviewer #2: Yes

2. Has the statistical analysis been performed appropriately and rigorously? 

Reviewer #1: Yes

Reviewer #2: Yes

3. Have the authors made all data underlying the findings in their manuscript fully available?

Reviewer #1: Yes

Reviewer #2: Yes

4. Is the manuscript presented in an intelligible fashion and written in standard English?

Reviewer #1: Yes

Reviewer #2: Yes

5. Review Comments to the Author

Reviewer #1: This manuscript is about the development of a qPCR to detect Phytophthora abietivora which is an emerging pathogen of Christmas tree (firs) in North America. The authors discuss the development of the assay in sufficient detail and discuss the results in the context of its future application as a monitoring and detection tool. The quality of the work in this manuscript appears sound. The qPCR assay is specific to the P. europaea complex so the authors need to be careful to state it being species specific when they themselves declare it semi specific. I would recommend this manuscript be accepted with minor changes.

Line 21 - Provide example of Phytophthora disease of a natural ecosystem which threatens biodiversity, or write something like “global plant biodiversity, horticulture and trade.”

Line 21 – species name for Christmas trees

Line 26 – remove the word generation

Line 27 – scientific name for firs

Line 32 – which species of firs were included?

Line 37 – how many of those eDNA samples were from symptomatic trees and did the results correlate with baiting? If it cross reacts with P. europaea, would you not report the result as positive for P. abietivora/P. europaea?

Line 47 – Reference 7 (O’Brien et al 2009) seems out of place here.

Line 49 – include (genus Abies) after true firs.

Line 54 – roots not root

Line 66 – species name for Fraser fir

Line 83 – species instead of agents

Line 191 – company details for hemocytometer?

Line 197 – garden soil collected from underneath what plants at what site? How did you know it was free of P. abievitora? Would there be other phytophthora present?

Line 199 – did you use the Powersoil pro kit for the extractions with soil?

Line 243 – so the qPCR (primers and probe) target P. abievitora/P. europaea/P. flexuosa/P.tyrrenhica/P. uliginosa? If both P. abievirora and P. europaea are associate with Christmas trees I think this limitation should be discussed

Line 260 – P. cinnamomic misspelled

Line 261 – amplified misspelled

Line 301 – arguably false positives can be more detrimental as it reduces the reliability and confidence in the testing method, unless positive detections are corroborated with another detection method (e.g. baiting)

Line 311 – include 0 in 0.931. Be careful with , in place of . depending on the journal.

Line 324 – can you please expand on this sentence “Of 488 soil samples, 68 were declared positive for P. abietivora.” How many of the 68 positives were from symptomatic trees and from plantations or natural stands? Were there any negative control soil samples? Were the positive results confirmed with soil baiting (from the previous study in which they were collected)? If so how many were positive with baiting vs this qPCR? Did any of the samples have P. europaea during baiting – i.e. is there any chance that any of the 68 positive trees were positive for P. europaea rather than P. abietivora?

Line 244, space missing here t(

Line 349 – space missing after P.

Line 382 – specific to P. abietivora and P. europaea? Semi-specific

Line 397 – italicise Phytophthora

Line 468 – culturing would also enable confirmation of species identification i.e. P. abievitora or P. europaea right?

General comments:

Figure 1 D, forest not Forest for title

Figure 3 A, x axis title more details, Proportion what

Figure 3 B x axis capital letter missing in Probability

Reviewer #2: The study presents the development of a sensitive and specific qPCR assay targeting Phytopthora abietivora and validating its efficiency on root and in soil samples. The qPCR assay will prove useful in early detection of the pathogen in Christmas tree plantations and of crucial information o the distribution of this emerging species in nature and of a better understanding of its ecology.

The ms is well written and each step of the qPCR assay clearly described. The results are presented with adequate statistical methods and described in sufficient detail.

General comments:

The samples used are describes as eDNA , however, environmental DNA or eDNA describes the genetic material present in environmental samples such as sediment, water, and air, including whole cells, extracellular DNA and potentially whole organisms. It will be more plain if you explain soil and root samples, eg in the abstract and elsewhere.

Introduction:

The introduction is very consise and I really miss information about the economic importance of Ornamental plants, and of the estimated loss due to PRR. There is an informative introduction in your article published earlier this year (Survey of Phytopthora Diversity Reveals P. abietivora as a Potential Phytopthora Root Rot Pathogen in Québec Christmas Tree Plantations). Each article must be read independently, so I will appreciate if you can add more information both in the introduction and in the discussion.

Page 4

Line 17. Unclear sentence.” In environmental or diseased tissues”

Page 6

Line 7 in S1 Table. Please correct.

Page 9

Line 6. Please explain what you refer to using environmental samples

Page 10

Line 5. Without soil? Unclear

Line 7. Please describe the origins of the garden soil.

Line 17. A brief recapitulation of the sampling procedure of soil cores will improve the understanding

Line 21. The plantations sampled were of different age?

PAGE 11.

Line 4. Please add the information on storing the samples prior to analysis.

Discussion:

Did you find any differences between different ages of plantations?

6. PLOS authors have the option to publish the peer review history of their article (what does this mean? ). If published, this will include your full peer review and any attached files.

**Do you want your identity to be public for this peer review?** For information about this choice, including consent withdrawal, please see our Privacy Policy .

Reviewer #1: No

Reviewer #2: No

---

## [Author Response · Author response to Decision Letter 1]

5 Feb 2025

All of the responses have been included in the "response to reviewer" file.

---

## [Decision Letter · Decision Letter 1]

24 Feb 2025

A qPCR assay for the detection of *Phytophthora abietivora* , an emerging pathogen on fir species cultivated as Christmas trees

PONE-D-24-40058R1

Dear Dr. Charron,

We’re pleased to inform you that your manuscript has been judged scientifically suitable for publication and will be formally accepted for publication once it meets all outstanding technical requirements.

Kind regards,

Kandasamy Ulaganathan

Academic Editor

PLOS ONE

Additional Editor Comments (optional):

Reviewers' comments:

Reviewer's Responses to Questions

**Comments to the Author**

1. If the authors have adequately addressed your comments raised in a previous round of review and you feel that this manuscript is now acceptable for publication, you may indicate that here to bypass the “Comments to the Author” section, enter your conflict of interest statement in the “Confidential to Editor” section, and submit your "Accept" recommendation.

Reviewer #1: All comments have been addressed

2. Is the manuscript technically sound, and do the data support the conclusions?

Reviewer #1: Yes

3. Has the statistical analysis been performed appropriately and rigorously? 

Reviewer #1: Yes

4. Have the authors made all data underlying the findings in their manuscript fully available?

Reviewer #1: Yes

5. Is the manuscript presented in an intelligible fashion and written in standard English?

Reviewer #1: Yes

6. Review Comments to the Author

Reviewer #1: The authors have addressed all suggested revisions and improved the quality of the manuscript. I recommend it be accepted for publication. I noticed a couple of minor mistakes such as italicised brackets, but these are only grammatical and will be picked up in the final proof. The authors have clearly explained and discussed any limitations (such as the specificity of the assay) and they have added details where requested (particularly in the methods section).

7. PLOS authors have the option to publish the peer review history of their article (what does this mean? ). If published, this will include your full peer review and any attached files.

**Do you want your identity to be public for this peer review?** For information about this choice, including consent withdrawal, please see our Privacy Policy .

Reviewer #1: No

---

## [Editor Report · Acceptance letter]

PONE-D-24-40058R1

PLOS ONE

Dear Dr. Charron,

I'm pleased to inform you that your manuscript has been deemed suitable for publication in PLOS ONE. Congratulations! Your manuscript is now being handed over to our production team.

Kind regards,

on behalf of

Dr. Kandasamy Ulaganathan

Academic Editor

PLOS ONE